# Predicting hypotension, syncope, and fracture risk in patients indicated for antihypertensive treatment: the STRATIFY models

Constantinos Koshiaris [1,2] ✉, Ariel Wang [1], Lucinda Archer[3,4],
Richard D. Riley[3,4], Kym IE Snell[3,4], Richard J. Stevens [1], Amitava Banerjee [5],
Subhashisa Swain[1], Andrew Clegg[6], Christopher E. Clark [7], Rupert A. Payne [7],
FD Richard Hobbs [1], Margaret Ogden[8], Richard J. McManus [1,9] &
James P. Sheppard [1] ✉ On behalf of the STRATIFY investigators*

Antihypertensives are associated with increased risk of syncope, hypotension, and fractures, but the highest-risk individuals are unclear. This study aimed to develop and validate three models to predict these outcomes in patients with an indication for antihypertensive treatment. A cohort study was conducted using data from Clinical Practice Research Datalink (CPRD). Patients aged 40+ with systolic blood pressure 130-179 mmHg were included. Outcomes were first hypotension, syncope, or fracture leading to hospitalization or death within 10 years. Models were derived from CPRD GOLD data ($n = 1,773,224$) and validated with CPRD Aurum data ($n = 3,805,366$). Each model had 31-37 predictors. Validation demonstrated strong discriminative ability (10-year C-statistic: hypotension 0.824; syncope 0.819; fracture 0.790), with close agreement between predicted and observed risks for the hypotension and syncope models. Some underprediction was observed for the fracture model. These models could be used to help reassure patients about the relatively low risk of harm from antihypertensive treatment, or identify the small number of individuals with a higher risk where additional monitoring may be indicated.

Hypertension is the leading risk factor for cardiovascular disease, making it an important target for intervention in routine clinical practice[1]. Blood pressure-lowering with antihypertensive treatment has been shown to be very effective at reducing the risk of cardio-vascular events across all age groups[2]. However, blood pressure lowering is not without harm[3]. Previous studies have highlighted the increased risk of adverse events such as hypotension, syncope, falls, acute kidney injury, and electrolyte abnormalities, especially in older patients and those with frailty[4]. For some individuals, where the risk of harm is high, it may not be appropriate to prescribe antihypertensive

[1]Nuffield Department of Primary Care Health Sciences, University of Oxford, Oxford, UK. [2]Department of Primary Care and Population Health, University of Nicosia Medical School, Nicosia, Cyprus. [3]Department of Applied Health Sciences, The University of Birmingham, Birmingham, UK. [4]National Institute for Health and Care Research (NIHR) Birmingham Biomedical Research Centre, Birmingham, UK. [5]Institute of Health Informatics, University College London, London, UK. [6]Academic Unit for Ageing and Stroke Research, Bradford Institute for Health Research, University of Leeds, Leeds, UK. [7]Exeter Collaboration for Academic Primary Care, University of Exeter Medical School, St Luke's Campus, Magdalen Rd, Exeter, UK. [8]Patient and Public Advisor, Durham, UK. [9]Brighton and Sussex Medical School University of Brighton and University of Sussex Brighton, Brighton, UK. *A list of authors and their affiliations appears at the end of the paper. ✉e-mail: koshiaris.c@unic.ac.cy; james.sheppard@phc.ox.ac.uk

treatment. In those already treated, interventions such as deprescribing may be considered[5,6]. To enable informed decision-making, clinicians need to understand an individual's underlying risk of adverse events, so that this can be weighed against a patient's likelihood of benefit from new or continued treatment.

In previous studies, clinical prediction models have been developed to estimate the risk of serious falls and acute kidney injury in patients indicated for antihypertensive therapy[7,8]. Although most patients in these studies with a high risk of acute kidney injury or falls also had a high risk of cardiovascular disease (CVD) a small number were shown to be at high risk of adverse events but low risk of cardiovascular disease[8]. In these individuals, new or continued treatment may not be appropriate. Using prediction models to understand an individual's risk of specific adverse events, treatment strategies can be personalised to ensure antihypertensive treatment is only prescribed to those with the most to gain.

In this study, we develop and externally validate three clinical prediction models for adverse events commonly associated with antihypertensive treatment—namely hypotension, syncope, and fracture—using data from over five million patients in the Clinical Practice Research Datalink. These models estimate baseline risk regardless of treatment status and are designed to support personalised prescribing decisions by identifying individuals at higher risk of adverse outcomes.

## Results

### Study population characteristics

A total of 1,773,224 patients were included in the model development cohort (CPRD GOLD) with a mean age of 59 years (SD 13 years) and a mean systolic blood pressure at study inclusion of 144 mmHg (SD 12 mmHg) (Table 1, figure S1). The 10-year prevalence of hazardous hypotension was 1.6% ($n = 28,450$), syncope was 2.2% ($n = 39,898$), and fracture was 4.1% ($n = 73,491$). The median follow-up time across the whole cohort was 6 years (IQR 2.6 to 10 years).

In the validation cohort, 3,805,366 patients were included, with 63,019 (1.7%) experiencing a hazardous hypotensive event, 84,262 (2.2%) a syncope event and 151,630 (4.0%) a fracture event during 10-year follow-up (table S3). Median follow-up time in the validation cohort was 7.0 years (2.9 to 10). Ethnicity data were more complete in the validation cohort compared to the development cohort (81% vs 44% complete data).

### Model Development

A total of 31 predictors were included in the final STRATIFY-Hypotension model, after the exclusion of covariates with little or no association with hypotension (Table 2). High social deprivation, current smoking status, previous hypotension, chronic kidney disease, and Parkinson's disease were strong predictors of hypotension resulting in hospitalisation or death. Prescription of all types of antihypertensive medication were associated with an increased risk of hazardous hypotension, with ACE inhibitors (SHR 1.41, 95% CI 1.37 to 1.45), angiotensin II receptor antagonists (SHR 1.36, 95% CI 1.30 to 1.43) and alpha blockers (SHR 1.35, 95% CI 1.26 to 1.45) conferring the greatest risk.

A total of 31 predictors were included in the final STRATIFY-Syncope model (Table 2). Covariates predictive of syncope were similar to those predictive of hypotension, with the exception of South Asian ethnicity, other ethnicity, dementia and heart failure which were all associated with a reduced risk of syncope requiring hospitalisation or leading to death, and antipsychotic medication prescription which was associated with increased risk.

A total of 37 predictors were included in the final STRATIFY-Fracture model (Table 2). Of these, 14 were unique predictors of fracture that were not included in the STRATIFY-Hypotension and STRATIFY-Syncope models. Strong predictors of fracture included heavy drinking, female sex, chronic liver disease, previous fracture, multiple sclerosis, epilepsy, osteoporosis and rheumatoid arthritis. All

antihypertensive medications had a weak or no association with the risk of fracture. Other medications were associated with an increased risk of fracture, with the exception of hormone replacement therapy which conferred a lower risk of fracture (Table 2).

Age was not linearly related with any of the outcomes so transformations were used. Miscalibration was observed across all models at 5 and 10 years so they were recalibrated to the observed pseudo-values in the development dataset (figure S2).

### External validation

The distribution of the prognostic index for the derivation and external validation datasets can be seen in the appendix (figure S3), and model performance statistics are given in Table 3, S4 and S5 and Fig. 1. The final STRATIFY-Hypotension model exhibited strong discriminative ability at 10 years (C-statistic 0.824, 95% CI 0.823 to 0.826) and close agreement between predicted and observed risks depending on time horizon (Observed/Expected [O/E] at 10 years 0.983, 95% CI 0.961 to 1.005). The STRATIFY-Syncope model also showed strong discriminative ability (C-statistic at 10 years 0.819, 95% CI 0.817 to 0.821) and close agreement between predicted and observed risks (O/E ratio at 10 years 1.028, 95% CI 1.009 to 1.047). The STRATIFY-Fracture model showed good discrimination (C-statistic at 10 years 0.790, 95% CI 0.789 to 0.792) and close agreement between predicted and observed risks but with some underprediction for low probabilities (O/E ratio at 10 years 1.13, 95% CI 1.11 to 1.14). Model performance varied more among smaller practices, with more consistent performance seen as practice size increased (figures S4 to S6).

Using a threshold of 5% across all models 732,598 (41%) of the patients were classified as high risk for at least one of the three adverse events at 10 years. Amongst the patients who were classified as high risk for at least one adverse event, 280,326 (38%) were classified as high risk for all three, 287,345 (39%) were classified high risk only for fracture, 31,289 (4.3%) only for syncope and 2,829 (0.4%) only for hypotension. -24,195 (3.3%) were at high risk for both syncope and hypotension (Fig. 2). Results were similar when using a 10% threshold to define high risk patients in each model (figure S7).

Decision curve analysis indicated that all three models had clinical utility across all three time points (Fig. 3). For example, using the STRATIFY-Hypotension model with a 10-year time horizon to guide decisions on prescribing would result in a higher net benefit compared to a "deprescribe/don't treat anyone" strategy, and the same was true for the STRATIFY-Syncope model and the STRATIFY-Fracture model.

Subgroup analyses of the 10-year risk models showed similar performance in younger (< 65 years) and older patients (≥ 65 years) and in females and males (figures S8, S10; tables S6 and S7). There was some evidence of under-prediction of hypotension risk in patients of white, black and South Asian ethnicity (figures S11, S12, table S8), although net benefit was consistent across ethnic minority groups for all three models (figure S13).

### Comparison with CVD risk

When using a 10% risk threshold for both cardiovascular disease and adverse events, no patients had a high risk of adverse events but low risk of cardiovascular disease. At the 5% threshold, among those patients with a low risk of cardiovascular disease at 10 years, 244 (0.01%) had a high risk of hypotension, 2656 (0.2%) had a high risk of syncope and 17,040 (1%) had a high-risk fracture. Most patients had a high risk of cardiovascular disease but low risk adverse events (Fig. 4).

## Discussion

This study developed three clinical prediction models for adverse events related to antihypertensive treatment, which estimate the baseline risk of hypotension, syncope and fracture over the next 1, 5 and 10 years. The models demonstrated good discrimination and suggested that individuals were most likely to be classified at high risk

**Table 1 | Baseline characteristics of patients in the development dataset (CPRD Gold)**

| Variable | Total (N = 1,773,224) | Hypotension (n = 28,450) | Death from causes other than hypotension (competing risk) - (n = 191,765) | Syncope (n = 39,898) | Death from causes other than Syncope (competing risk) (n = 191,434) | Fracture (n = 73,491) | Death from causes other than Fracture (competing risk) (n = 182,131) |
|---|---|---|---|---|---|---|---|
| **Age, years – mean (SD)** | 59.4 (13.2) | 72 (12) | 75 (12) | 70 (13) | 75 (12) | 70.0 (14.0) | 74.4 (12.0) |
| **Systolic blood pressure, mmHg – mean (SD)** | 143.5 (12.0) | 150 (13) | 150 (13) | 150 (13) | 150 (13) | 146.1 (12.7) | 146.9 (12.8) |
| **Diastolic blood pressure, mmHg – mean (SD)** | 83.8 (9.6) | 82 (10) | 82 (10) | 83 (9.9) | 82 (10) | 82.4 (9.8) | 81.7 (10.1) |
| **Follow up, years (p50, IQR)** | 6.2 (2.6-10) | 4.8 (2.3–7.3) | 3.8 (1.7–6.4) | 4.2 (1.9–6.8) | 3.7 (1.7–6.4) | 3.9 (1.7–6.6) | 3.7 (1.6–6.3) |
| **Sex** | | | | | | | |
| Male | 851,058 (48%) | 14,002 (49.2%) | 92,552 (48.3%) | 19,620 (49.2%) | 92,696 (48.4%) | 24,131 (32.8%) | 91,838 (50.4%) |
| Female | 922,166 (52%) | 14,448 (50.8%) | 99,213 (51.7%) | 20,278 (50.8%) | 98,738 (51.6%) | 49,360 (67.2%) | 90,293 (49.6%) |
| **Ethnicity** | | | | | | | |
| White | 734,401 (41.4%) | 26,858 (94.4%) | 11,448 (59.7%) | 37,519 (94.0%) | 11,431431 (59.7%) | 69,403 (94.4%) | 105,826 (58.1%) |
| Black | 10,802 (0.6%) | 198 (0.7%) | 843 (0.4%) | 358 (0.9%) | 838 (0.4%) | 294 (0.4%) | 865 (0.5%) |
| South Asian | 14,805 (0.8%) | 361 (1.3%) | 998 (0.5%) | 413 (1.0%) | 1026 (0.5%) | 548 (0.7%) | 1021 (0.6%) |
| Other | 15,737 (0.9%) | 364 (1.3%) | 1271 (0.7%) | 469 (1.2%) | 1306 (0.7%) | 785 (1.1%) | 1212 (0.7%) |
| Missing | 997,479 (56.3%) | 669 (2.4%) | 74,167 (38.7%) | 1139 (2.9%) | 73,949 (38.6%) | 2461 (3.3%) | 73,207 (40.2%) |
| **Deprivation Score** | | | | | | | |
| IMD 1 | 419,468 (23.7%) | 5,558558 (19.55%) | 37,448448 (19.6%) | 7,861861 (19.77%) | 37,503503 (19.66%) | 15,576 (21.2%) | 35,578 (19.5%) |
| IMD 2 | 406,916 (22.9%) | 6097 (21.4%) | 41,783 (21.8%) | 8701 (21.8%) | 41,691 (21.8%) | 16,359 (22.3%) | 39,687 (21.8%) |
| IMD 3 | 376,903 (21.3%) | 6057 (21.3%) | 41,451 (21.6%) | 8277 (20.7%) | 41,444 (21.6%) | 15,703 (21.4%) | 39,383 (21.6%) |
| IMD 4 | 313,707 (17.7%) | 5585 (19.6%) | 37,040 (19.3%) | 733 (19.4%) | 36,972 (19.3%) | 13,827 (18.8%) | 35,206(19.3%) |
| IMD 5 | 254,800 (14.4%) | 5127 (18.0%) | 33,856 (17.7%) | 7280 (18.2%) | 33,639 (17.6%) | 11,972 (16.3%) | 32,096 (17.6%) |
| Missing | 1,430 (0.1%) | 26 (0.09%) | 187 (0.1%) | 46 (0.12%) | 185 (0.1%) | 54 (0.1%) | 181 (0.1%) |
| **BMI** | | | | | | | |
| Underweight | 20,635 (1.2%) | 584 (2.1%) | 5711 (3.0%) | 746 (1.9%) | 5695 (3.0%) | 2065 (2.8%) | 5085 (2.8%) |
| Normal | 519,524 (29.3%) | 8794 (30.9%) | 57,136 (29.8%) | 12,218 (30.6%) | 56,967 (29.8%) | 25,024 (34.0%) | 53,391 (29.3%) |
| Overweight | 586,531 (33.1%) | 8905 (31.3%) | 51,537 (26.9%) | 12,707 (31.8%) | 51,475 (26.9%) | 20,888 (28.4%) | 49,902 (27.4%) |
| Obese | 340,357 (19.2%) | 4788 (16.8%) | 25,528 (13.3%) | 6443 (16.1%) | 25,747 (13.4%) | 9945 (13.5%) | 25,170 (13.8%) |
| Morbidly obese | 39,853 (2.2%) | 551 (1.9%) | 2,728 (1.4%) | 618 (1.5%) | 2803 (1.5%) | 881 (1.2%) | 2,771 (1.5%) |
| Missing | 266,324 (15%) | 4828 (17.0%) | 49,125 (25.6%) | 7166 (18.0%) | 48,747 (25.5%) | 14,688 (20.0%) | 45,812 (25.2%) |
| **Smoking status** | | | | | | | |
| Non smoker | 847,473 (47.8%) | 12,107 (42.6%) | 79,576 (41.5%) | 18,843 (47.2%) | 78,886 (41.2%) | 35,461 (48.2%) | 74,443 (40.9%) |
| Ex-smoker | 471,193 (26.6%) | 8,907 (31.3%) | 53,345 (27.8%) | 10,817 (27.1%) | 53,716 (28.1%) | 18,349 (25%) | 51,665 (28.4%) |
| Smoker | 363,579 (20.5%) | 5537 (19.5%) | 39,653 (20.7%) | 7,236 (18.1%) | 39,852 (20.8%) | 13,514 (18.4%) | 38,359 (21.1%) |
| Missing | 90,979 (5.1%) | 1899 (6.7%) | 19,191 (10.0%) | 3002 (7.5%) | 18,980 (9.9%) | 6167 (8.4%) | 17,664 (9.7%) |
| **Alcohol** | | | | | | | |
| Non drinker | 289,581 (16.3%) | 6339 (22.3%) | 40,162 (20.9%) | 8359 (21.0%) | 40,074 (20.9%) | 14,701 (20.0%) | 37,764 (20.7%) |
| Trivial drinker | 488,448 (27.5%) | 7322 (25.7%) | 44,664 (23.3%) | 10,481 (26.3%) | 44,620 (23.3%) | 18,818 (25.6%) | 42,610 (23.4%) |
| Light drinker | 239,799 (13.5%) | 3091 (10.9%) | 19,539 (10.2%) | 4,455 (11.2%) | 19,572 (10.2%) | 8396 (11.4%) | 18,911 (10.4%) |
| Moderate drinker | 179,162 (10.1%) | 2078 (7.3%) | 13,196 (6.9%) | 2988 (7.5%) | 13,247 (6.9%) | 5282 (7.2%) | 12,950 (7.1%) |
| Heavy drinker | 22,772 (1.3%) | 417 (1.5%) | 2,414 (1.3%) | 589 (1.5%) | 2440 (1.3%) | 1074 (1.5%) | 2356 (1.3%) |
| Unknown amount | 291,767 (16.5%) | 4439 (15.6%) | 27,649 (14.4%) | 6261 (15.7%) | 27,585 (14.4%) | 11,548 (15.7%) | 26,232 (14.4%) |
| Missing | 261,695 (14.8%) | 4764 (16.7%) | 44,141 (23.0%) | 6765 (17.0%) | 43,896 (22.9%) | 13,672 (18.6%) | 41,308 (22.7%) |

**Table 1 (continued) | Baseline characteristics of patients in the development dataset (CPRD Gold)**

| Variable | Total (N = 1,773,224) | Hypotension (n = 28,450) | Death from causes other than hypotension (competing risk) - (n = 191,765) | Syncope (n = 39,898) | Death from causes other than Syncope (competing risk) (n = 191,434) | Fracture (n = 73,491) | Death from causes other than Fracture (competing risk) (n = 182,131) |
|---|---|---|---|---|---|---|---|
| **Antihypertensive drugs** | | | | | | | |
| ACE inhibitors | 219,588 (12.4%) | 7101 (25.0%) | 39,687 (20.7%) | 8217 (20.6%) | 40,034 (20.9%) | 11,667 (15.9%) | 38,554 (21.2%) |
| Angiotensin II receptor antagonists | 59,103 (3.3%) | 1788 (6.3%) | 8055 (4.2%) | 1951 (4.9%) | 8136 (4.3%) | 2920 (4.0%) | 7797 (4.3%) |
| Alpha blockers | 34,349 (1.9%) | 1405 (4.9%) | 7054 (3.7%) | 1635 (4.1%) | 7075 (3.7%) | 1985 (2.7%) | 6889 (3.8%) |
| Beta blockers | 216,202 (12.2%) | 6361 (22.4%) | 32,595 (17.0%) | 7765 (19.5%) | 32,758 (17.1%) | 11,141 (15.2%) | 31,573 (17.3%) |
| Calcium channel blockers | 193,221 (10.9%) | 6091 (21.4%) | 37,703 (19.7%) | 7729 (19.4%) | 37,548 (19.6%) | 11,614 (15.8%) | 35,965 (19.7%) |
| Diuretics | 180,115 (10.2%) | 5000 (17.6%) | 31,576 (16.5%) | 6900 (17.3%) | 31,372 (16.4%) | 11,638 (15.8%) | 29,687 (16.3%) |
| Other antihypertensives | 10,884 (0.6%) | 730 (2.6%) | 4860 (2.5%) | 751 (1.9%) | 4934 (2.6%) | 1262 (1.7%) | 4685 (2.6%) |

of a fracture, followed by syncope and hypotension. Calibration showed close agreement between predicted and observed risks for all models across all time horizons, with the exception of the hypotension model at 1 year, which tends to underestimate the risk. Some minor underprediction was also observed for the fracture model at 10 years for low predicted probabilities. Given this, we would suggest caution in using the 1-year hypotension model at this stage.

When compared to risk of cardiovascular disease, only a very small proportion of patients ( < 1%) were found to be at high risk of adverse events and low risk of cardiovascular disease. A large proportion of patients (39-56%) were identified to have high risk of cardiovascular disease and low risk of adverse events. This suggests that for the majority of people, the potential benefits of treatment will outweigh the risk of hypotension, syncope or fracture. The information from these models may therefore be useful in helping patients make informed decisions about their treatment options, potentially reducing unnecessary worry or apprehension.

As part of the clinical utility assessment, the predicted risks from each model were compared with the risk of cardiovascular disease, generated by the QRisk2 algorithm for specific thresholds[9]. This latter model was recommended by NICE during the study period and estimates the risk of cardiovascular disease in patients aged 35-84 years over a 10-year period, but does not consider the competing risk of death[10]. This can lead to overestimation of cardiovascular risk, particularly over shorter timeframes and in older patients with multiple health conditions where the competing risk of death from other causes is higher[11]. As a result, analyses showing that the risk of cardiovascular disease outweighs the risk of adverse events should be interpreted with caution, particularly over shorter timeframes.

Of the adverse events examined in the present study, fracture has most commonly been studied in previous risk prediction modelling studies[12]. Common examples include the FRAX score, Garvan Fracture Risk Calculator and the QFracture tool[13–15]. These typically focus on hip and osteoporotic fracture (FRAX and QFracture) and display varying performance upon external validation due to differences in underlying population and input variables[12,14,15]. However, unlike the present STRATIFY models, none of these previous models takes into account the competing risk of death and this has been shown to lead to significant over-prediction of fracture risk in older patients with multimorbidity[16]. This is important when considering adverse event risk in particular, where one treatment strategy for high risk patients might include deprescribing, or not starting therapy which still carries benefit. Few studies have examined the risk of hypotension or syncope, but these tend to focus on risk prediction during emergency department admission and inpatient stays in hospital[17–20]. One study examined the risk of postural hypotension in the community and found

moderate discrimination, but this model was not externally validated and calibration was not assessed[21].

Clinical guidelines for the management of hypertension are increasingly recommending consideration of deprescribing antihypertensive therapy in specific circumstances, where the benefits of treatment may be outweighed by the harms[22]. In the UK, the National Institute for Health and Care Excellence currently advises that clinicians should use clinical judgement in blood pressure lowering treatment decisions in the presence of multimorbidity[10]. The purpose of developing these risk prediction models was to help clinicians estimate the baseline risk of adverse reactions. The models can be applied to both patients that are on antihypertensive treatment or patients for which treatment is being considered. Clinicians can then combine the estimated baseline risk with relative treatment effects obtained from well conducted randomised clinical trials or observational studies to estimate how the risk gets modified by starting, changing or altering the dosage of a medication[3,4]. Based on this updated risk different treatment strategies can be considered depending on the outcome of interest. For example, in patients at high risk of hypotension and fracture, modification of antihypertensive treatment may be considered, whereas in patients at risk of fracture alone, other prevention strategies may be more appropriate. These tools should be used alongside CVD risk estimation tools to get a more complete picture of the harm/benefit profile of the patient. This can enable better informed decisions regarding when to prescribe, continue or deprescribe antihypertensive treatment. To this end, these algorithms could easily be integrated into electronic health records systems to work alongside existing risk stratification tools such as QRisk[9].

All three models suggested net clinical benefit when compared to usual care (with the exception of the 1-year hypotension model), which typically would not involve modifying treatment to account for adverse event risk. However, very few patients who were at high risk of serious hypotension or syncope were also observed to be at low risk of cardiovascular disease (between 0.01% and 1%), thus implementing interventions which withhold or deprescribe treatment due to the risk of hypotension or syncope alone is only likely to be considered for a very small number of patients.

Slightly more individuals were at high risk of fracture and low risk of cardiovascular disease (1%) when using a fracture and cardiovascular risk threshold of 5%, however, the direct association between antihypertensive treatment and fractures is disputed and is likely to be small[3,4,23,24]. Therefore enthusiasm for intervening in such patients should be tempered by a likely small effect from withholding or deprescribing treatment. For most patients, these models should be employed to provide reassurance that the risk of adverse events is low for the vast majority of the population even after taking into

**Table 2 | STRATIFY prediction models for Hypotension, Syncope and Fracture. Values represent sub-distribution hazard ratios and 95% confidence intervals**

| Variable | STRATIFY-Hypotension model | STRATIFY-Syncope model | STRATIFY-Fracture model |
|---|---|---|---|
| **Age** | 31.06 (29.39–32.81)* | 23.54 (22.48–24.65)* | 1.03 (1.03–1.03)* |
| **Sex (Female)** | 0.79 (0.77–0.81) | 0.77 (0.75–0.79) | 1.59 (1.57–1.62) |
| **Systolic blood pressure** | 1.53 (1.30–1.80)* | - | 1.00 (1.00–1.00) |
| **Ethnicity (ref. White)** | | | |
| Black | - | 0.98 (0.79–1.23) | 0.44 (0.35–0.56) |
| South Asian | - | 0.74 (0.64–0.85) | 0.62 (0.55–0.69) |
| Other | - | 0.84 (0.72–0.97) | 0.76 (0.68–0.86) |
| **Deprivation Score (ref. IMD 1)** | | | |
| IMD 2 | 1.06 (1.02–1.1) | 1.1 (1.06–1.13) | 1.05 (1.03–1.07) |
| IMD 3 | 1.11 (1.07–1.15) | 1.11 (1.08–1.15) | 1.08 (1.05–1.11) |
| IMD 4 | 1.21 (1.17–1.25) | 1.25 (1.21–1.29) | 1.17 (1.14–1.19) |
| IMD 5 (High) | 1.32 (1.27–1.37) | 1.43 (1.39–1.47) | 1.24 (1.21–1.27) |
| **BMI (ref. Normal)** | | | |
| Underweight | - | - | 1.25 (1.19–1.31) |
| Overweight | - | - | 0.84 (0.83–0.86) |
| Obese | - | - | 0.75 (0.73–0.77) |
| Morbidly obese | - | - | 0.66 (0.62–0.70) |
| **Smoking status (ref. Non smoker)** | | | |
| Ex-smoker | 1.24 (1.2 – 1.27) | - | 1.01 (0.99 – 1.03) |
| Smoker | 1.42 (1.37 – 1.47) | - | 1.16 (1.13 – 1.18) |
| **Alcohol (ref. Non drinker)** | | | |
| Trivial drinker | 0.89 (0.86–0.92) | 0.91 (0.88–0.94) | 0.97 (0.95–0.99) |
| Light drinker | 0.84 (0.81–0.88) | 0.86 (0.83–0.9) | 1.02 (0.99–1.05) |
| Moderate drinker | 0.84 (0.80–0.88) | 0.86 (0.82–0.89) | 1.13 (1.10–1.17) |
| Heavy drinker | 1.29 (1.17–1.43) | 1.32 (1.2–1.45) | 1.79 (1.68–1.92) |
| Unknown amount | 0.9 (0.86–0.93) | 0.92 (0.89–0.95) | 1.01 (0.98–1.03) |
| **Risk Factors** | | | |
| Dizziness | 1.15 (1.11–1.19) | 1.15 (1.12–1.18) | - |
| Dementia | - | 0.72 (0.67–0.78) | 0.74 (0.71–0.78) |
| Multiple sclerosis | - | - | 1.41 (1.29–1.55) |
| Hypotension | 1.94 (1.83–2.07) | 1.32 (1.24–1.42) | - |
| Syncope | 1.4 (1.32–1.49) | 2.21 (2.13–2.3) | - |
| Previous Falls | 1.11 (1.07–1.16) | 1.18 (1.15–1.21) | 1.22 (1.19–1.25) |
| Previous Fracture | - | - | 1.57 (1.54–1.60) |
| Stroke | 1.06 (1.01–1.11) | 1.28 (1.23–1.33) | 1.06 (1.02–1.10) |
| Heart failure | - | 0.88 (0.83–0.92) | - |
| Chronic kidney disease | 1.46 (1.38–1.55) | - | 0.93 (0.89–0.97) |
| Diabetes | 1.27 (1.22–1.31) | 1.25 (1.22–1.29) | 1.22 (1.18–1.25) |
| Parkinson's disease | 1.72 (1.62–1.81) | 1.17 (1.09–1.26) | 1.29 (1.22–1.36) |
| Spinal cord injury | - | - | - |
| Ischaemic heart disease | 1.21 (1.17–1.24) | 1.26 (1.23–1.29) | - |
| Atrial fibrillation | 1.15 (1.1–1.21) | 1.06 (1.02–1.11) | - |
| Anaemia | 1.12 (1.08–1.17) | - | - |
| Bradycardia | 1.13 (1.01–1.26) | 1.12 (0.99–1.27) | - |

**Table 2 (continued) | STRATIFY prediction models for Hypotension, Syncope and Fracture. Values represent sub-distribution hazard ratios and 95% confidence intervals**

| Variable | STRATIFY-Hypotension model | STRATIFY-Syncope model | STRATIFY-Fracture model |
|---|---|---|---|
| Tachycardia | 1.4 (1.29–1.52) | 1.19 (1.1–1.29) | - |
| Structural cardiac disease | 1.22 (1.17–1.27) | 1.18 (1.14–1.22) | - |
| Cardiopulmonary disease | 1.31 (1.21–1.42) | 1.17 (1.08–1.26) | - |
| Osteoporosis | - | - | 1.30 (1.25–1.34) |
| Rheumatoid Arthritis | - | - | 1.30 (1.24–1.37) |
| Gastrointestinal Conditions | - | - | 1.14 (1.08–1.21) |
| Epilepsy | - | - | 1.37 (1.29–1.46) |
| Respiratory problems | - | - | 1.06 (1.04–1.08) |
| Chronic liver disease | - | - | 1.62 (1.47–1.80) |
| **Anti-hypertensive drugs** | | | |
| ACE inhibitors | 1.41 (1.37–1.45) | 1.21 (1.17–1.24) | 1.05 (1.03–1.07) |
| Angiotensin II receptor antagonists | 1.36 (1.30–1.43) | 1.15 (1.1–1.21) | 1.04 (1.00–1.07) |
| Alpha blockers | 1.35 (1.26–1.45) | 1.24 (1.18–1.31) | 1.03 (0.98–1.09) |
| Beta blockers | 1.29 (1.25–1.33) | 1.15 (1.12–1.18) | 1.00 (0.98–1.02) |
| Calcium channel blockers | 1.19 (1.16 to 1.23) | 1.15 (1.12 to 1.18) | 1.06 (1.04–1.08) |
| Diuretics | 1.11 (1.07–1.15) | 1.15 (1.12–1.17) | 1.04 (1.02–1.06) |
| Other anti-hypertensives | 1.22 (1.13–1.33) | 1.08 (1–1.17) | 1.14 (1.07–1.21) |
| **Other drugs** | | | |
| Opioids | 1.30 (1.27–1.34) | 1.19 (1.16–1.23) | 1.16 (1.13–1.18) |
| Hypnotics, anxiolytics | 1.07 (1.03–1.11) | 1.12 (1.09–1.15) | - |
| Antipsychotics | - | 1.19 (1.13–1.26) | 1.12 (1.07–1.17) |
| Antidepressants | 1.31 (1.28–1.35) | 1.24 (1.2–1.28) | 1.19 (1.16–1.21) |
| Osteoporosis medications | - | - | 1.20 (1.16–1.24) |
| Systemic corticosteroids | - | - | 1.07 (1.04–1.09) |
| Hormone replacement therapy | - | - | 0.76 (0.74–0.78) |
| Anticonvulsants | - | - | 1.37 (1.29 to 1.44) |
| Proton pump inhibitors | - | - | 1.08 (1.06 to 1.10) |

*Variable transformed to account for non-linear association with the outcome
*IMD* Index of multiple deprivation; *BMI* Body mass index; *ACE* Angiotensin converting enzyme; *H2RA* Histamine type-2 receptor antagonists

consideration treatment effects from other studies. This information can be useful for clinicians and patients in helping them to make informed decisions about their treatment options, potentially reducing unnecessary apprehension about starting treatment. Where patients are considered at high risk of adverse events such as fracture, other strategies besides modifying antihypertensive treatment may be more appropriate such as monitoring or addressing some of the other risk factors that might be increasing the risk.

This analysis has several strengths, including the robust analytical approach with both internal and external validation across multiple practices, which demonstrate each of the models' reliability and

**Table 3 | Predictive performance statistics at 10 years for the final STRATIFY-Hypotension, STRATIFY-Syncope and STRATIFY-Fracture models upon external validation in CPRD Aurum**

| | STRATIFY-Hypotension | | STRATIFY-Syncope | | STRATIFY-Fracture | |
|---|---|---|---|---|---|---|
| | Original model (before re-calibration)* | Final model (after re-calibration) | Original model (before re-calibration)* | Final model (after re-calibration) | Original model (before re-calibration)* | Final model (after re-calibration) |
| **Observed/Expected** | | | | | | |
| Pooled effect size (95% CI) | 0.976 (0.955 to 0.998) | 0.983 (0.961 to 1.005) | 0.968 (0.938 to 1.000) | 1.028 (1.009 to 1.047) | 1.038 (1.024 to 1.052) | 1.13 (1.114 to 1.146) |
| 95% Prediction interval | 0.55 to 1.73 | 0.56 to 1.73 | 0.43 to 2.18 | 0.64 to 1.64 | 0.737 to 1.462 | 0.79 to 1.615 |
| $Tau^2$ | 0.084 (0.076 to 0.094) | 0.083 (0.074 to 0.092) | 0.170 (0.153 to 0.190) | 0.0549 (0.0511 to 0.0636) | 0.03 (0.027 to 0.034) | 0.033 (0.03 to 0.037) |
| **C statistic*** | | | | | | |
| Pooled effect size (95% CI) | 0.824 (0.823 to 0.826) | 0.824 (0.823 to 0.826) | 0.819 (0.817 to 0.821) | 0.819 (0.817 to 0.821) | 0.79 (0.789 to 0.792) | 0.79 (0.789 to 0.792) |
| 95% Prediction interval | 0.779 to 0.862 | 0.779 to 0.862 | 0.773 to 0.857 | 0.774 to 0.857 | 0.747 to 0.828 | 0.747 to 0.828 |
| $Tau^2$ | 0.021 (0.019 to 0.024) | 0.021 (0.019 to 0.024) | 0.020 (0.018 to 0.023) | 0.020 (0.018 to 0.023) | 0.016 (0.014 to 0.018) | 0.016 (0.014 to 0.018) |
| **D statistic** | | | | | | |
| Pooled effect size (95% CI) | 1.383 (1.273 to 1.493) | 1.359 (1.251 to 1.467) | 1.190 (1.096 to 1.285) | 1.174 (1.081 to 1.268) | 1.318 (1.306 to 1.329) | 1.317 (1.305 to 1.329) |
| 95% Prediction interval | 1.27 to 1.49 | 1.25 to 1.47 | 1.10 to 1.28 | 1.08 to 1.27 | 1.122 to 1.514 | 1.119 to 1.515 |
| $Tau^2$ | 0.001 (<0.001 to 0.015) | <0.001 (<0.001 to 0.015) | <0.001 (<0.001 to 0.012) | <0.001 (<0.001 to 0.012) | 0.01 (0.008 to 0.012) | 0.01 (0.008 to 0.013) |
| **Royston and Sauerbrei's $R^2_D$** | | | | | | |
| Range | 3.1 to 70.1 | 3.1 to 70.2 | 2.2 to 76.3 | 2.1 to 76.3 | 13.8 to 99.9 | 13.8 to 99.9 |
| Median (IQR) | 34.1 (29.8 to 39.0) | 33.6 (29.3 to 38.3) | 27.9 (24.5 to 32.0) | 27.6 (24.1 to 31.5) | 29.4 (26.5 to 32.7) | 29.3 (26.5 to 32.8) |
| Mean (SD) | 34.8 (8.3) | 34.2 (8.2) | 28.9 (7.6) | 28.3 (7.5) | 30 (6.6) | 30 (6.6) |

*Models re-calibrated in the development dataset, and then validated separately in the validation dataset.
**Pooled on natural log scale,
***Pooled on logit scale.

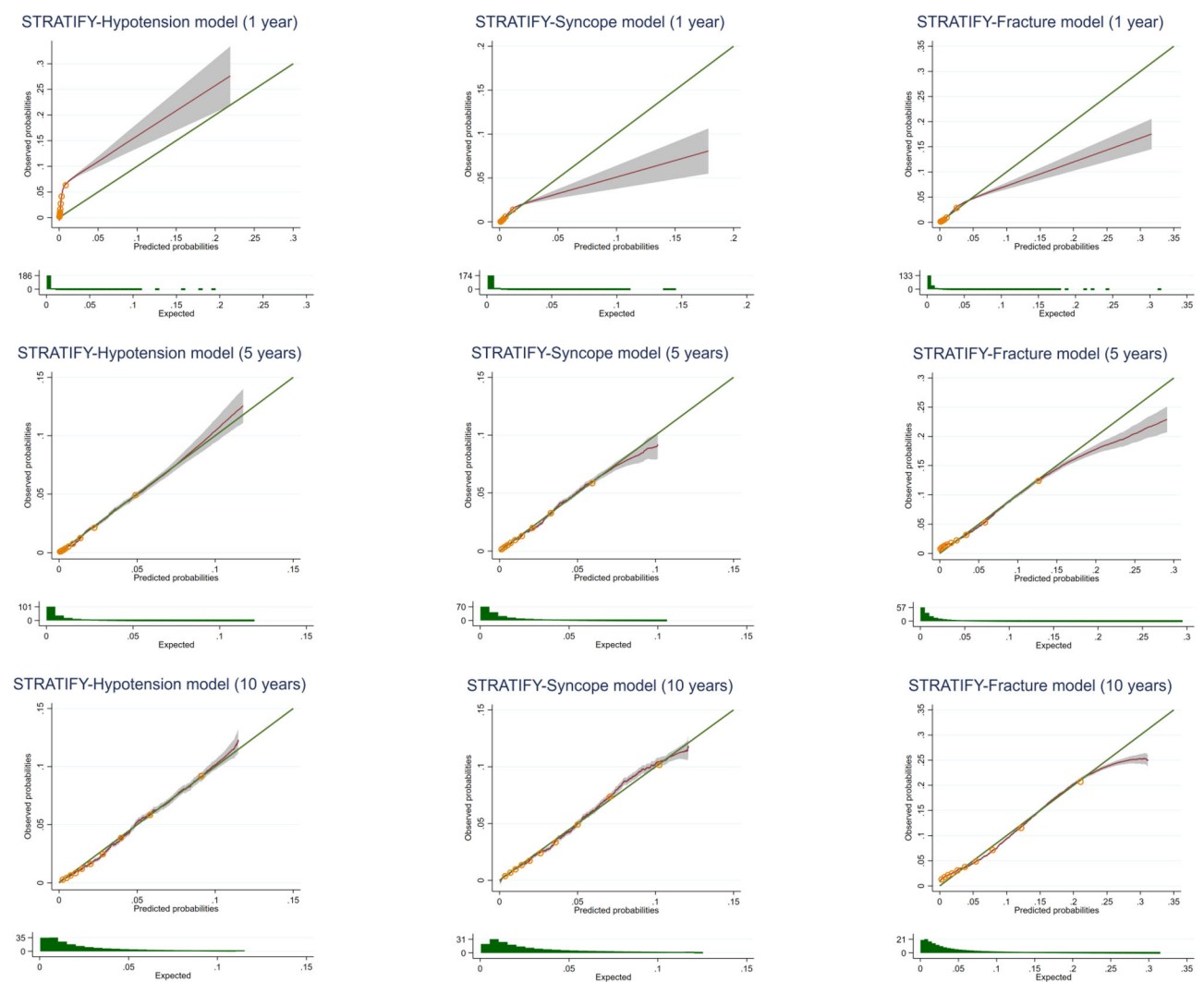

**Fig. 1 | Calibration curves for the external validation performance of the final STRATIFY models (CPRD Aurum).** Green line corresponds to the line of equality, red line the calibration curve with 95% CI and yellow dots the deciles of predicted risk. The 5- and 10-year models were re-calibrated using the derivation dataset. For one-year re-calibration was not required. Groups represent tenths of the linear predictor, as created between deciles. Histograms underneath each calibration plot show the distribution of predicted probabilities.

generalisability within the UK primary care population[25]. There are also some limitations. These data may not accurately capture all events of interest (due to incorrect or incomplete coding), potentially affecting the model's performance, especially if certain events are systematically underreported or misclassified[26]. Findings from clinical utility analyses should be interpreted with caution, as they estimate factual risk (i.e., risk based on baseline characteristics). The net benefit in Decision Curve Analysis (DCA) reflects the clinical utility of using the model's predictions for treatment decision-making, without accounting for the causal effect of altering antihypertensive medication on outcomes. This study primarily aims to predict baseline risk and does not evaluate how treatment changes will modify these risks. While the model enhances risk stratification and supports decision-making by identifying high- and low-risk patients, it does not provide direct insight into how starting or adjusting antihypertensive treatments affects patient outcomes. Thus, any conclusions regarding treatment effects should rely on additional evidence, such as randomised controlled trials or observational studies. However, it is reasonable to assume that a model with higher net benefit may still improve clinical outcomes by better targeting interventions

The present study used large datasets of electronic health records to derive and externally validate three clinical prediction models to estimate the baseline risk of adverse events associated with

antihypertensive therapy. These models were shown to perform well but revealed that only a small proportion of patients eligible for antihypertensive treatment are at high risk of adverse events in the short to medium term. Therefore, these models are most likely to be of clinical utility in providing reassurance to patients considering antihypertensive treatment, potentially reducing unnecessary worry or apprehension.

## Methods
### Ethics approval
The study protocol was approved by CPRD's Independent Scientific Advisory Committee in February 2019 before obtaining the data relevant to the project (protocol given in the eAppendix in the Supplement). All data are fully anonymised so consent was not required. A project summary is published on the CPRD website (https://www.cprd.com/isac). Elements of the methodology used in this manuscript have been previously reported in related publications from the same research programme, using the same dataset and protocol and are summarised here for completeness[7,8,27].

### Design
We conducted a retrospective observational cohort study to develop three clinical prediction models using data from the Clinical Practice

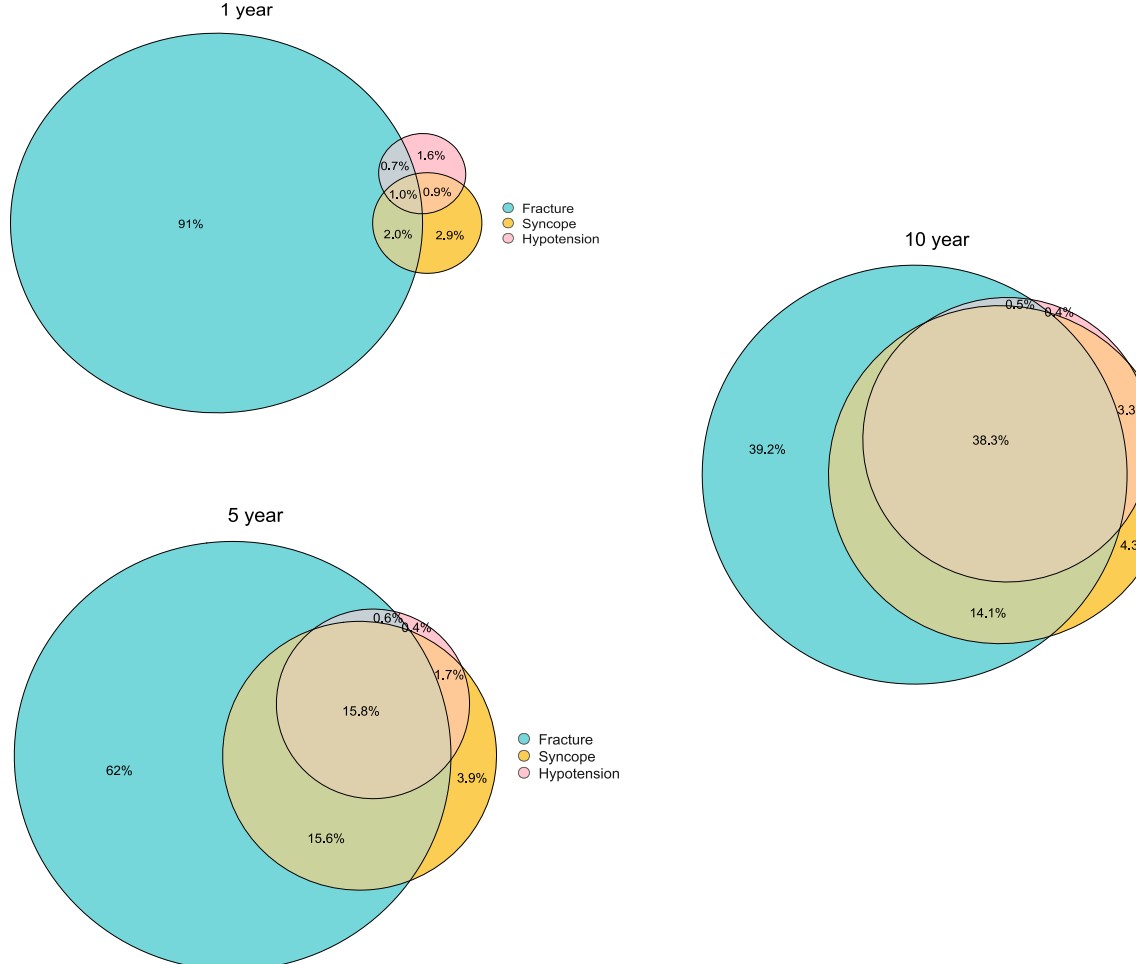

**Fig. 2 | Overlap of High-Risk Patients Identified by STRATIFY Models (CPRD Gold).** Venn diagrams showing the overlap of high-risk patients (≥ 5%) as classified using the final STRATIFY-Hypotension, STRATIFY-Syncope, and STRATIFY-Fracture in the CPRD Gold cohort (derivation). Denominator population is the total number of patients with either a high (≥ 5%) STRATIFY-Hypotension, STRATIFY-Syncope or STRATIFY-Fracture risk.

Research Datalink (CPRD) GOLD. This dataset includes primary care records from general practices that use the Vision electronic health record system (Cegedim Healthcare Solutions, London, England). The cohort comprised 11.33 million patients from 674 general practices, of whom 4.4 million were active (alive)[28]. For external validation, we used a second retrospective observational cohort based on CPRD Aurum, which contains data from practices using Egton Medical Information Systems (EMIS, Leeds, England) which t included 19 million patients from 738 practices, with 7 million active patients[29]. Both CPRD GOLD and Aurum datasets are representative of the UK population in terms of age, sex, ethnicity, and deprivation[28,29]. Primary care data from both sources were linked to additional datasets, including Office for National Statistics (ONS) mortality data, Hospital Episode Statistics (HES), and the Index of Multiple Deprivation (IMD). The study protocol was approved by the Independent Scientific Advisory Committee (ISAC) for CPRD (protocol number 19_042; see Protocol S1 in the supplementary material).

## Population

Patients were eligible for inclusion if they were registered at a general practice in England contributing linked data to CPRD GOLD between 1 January 1998 and 31 December 2018. To avoid duplicate entries, individuals appearing in both CPRD GOLD and CPRD Aurum due to transitions between electronic health record systems during the study period were excluded from the CPRD Aurum (validation) dataset. Inclusion criteria required patients to be aged 40 years or older at the time of data entry (with no upper age limit), registered with a CPRD "up-to-standard" practice (for GOLD only), and to have records available during the defined study period.

Patients entered the cohort at the point they became potentially eligible for antihypertensive therapy, defined as the date of their first systolic blood pressure measurement ≥130 mmHg following the start of the study period. Follow-up continued for a maximum of 10 years.

The 130 mmHg threshold was selected to align with the varying treatment initiation criteria outlined in international hypertension guidelines[22,30]. Patients with a systolic blood pressure measurement ≥180 mmHg were excluded from the cohort, as treatment would be indicated for them regardless of their estimated risk of adverse outcomes.

Baseline patient characteristics and model predictors were assessed at the index date, defined as 12 months after cohort entry. The same eligibility criteria and procedures for determining baseline characteristics were applied consistently to both the development and validation cohorts.

Patients exited the cohort at the end of follow-up (31 December 2018) or upon transferring out of a CPRD-registered practice, death, or occurrence of the specific outcome of interest.

## Outcomes

For each model, the primary outcome was defined as any hospitalisation or death with a primary diagnosis of (hazardous) hypotension, syncope, or fracture occurring within 10 years of the index date. This 10-

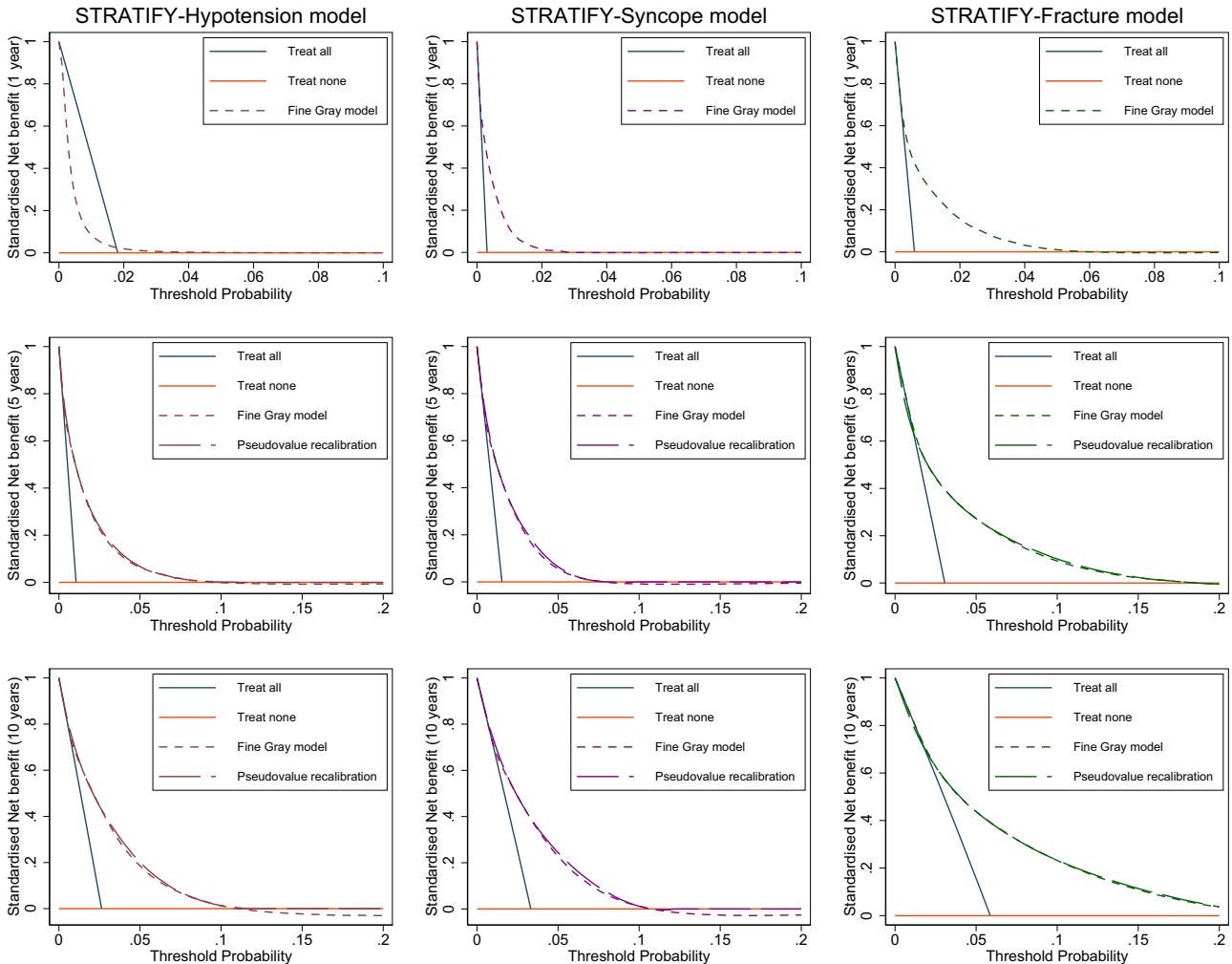

**Fig. 3 | Decision curve analysis of STRATIFY Models (CPRD Aurum).** Decision curves, showing the smoothed, standardised net benefit of using the prediction models across different threshold probabilities for assigning treatment. Treat all corresponds to introducing adverse event prevention measures (which may include deprescribing) for all patients and treat none corresponds to not introducing adverse event prevention measures for all patients.

year time horizon aligns with those used in established cardiovascular risk prediction models[9]. Outcomes were identified using ICD-9 and ICD-10 codes recorded in Hospital Episode Statistics (HES) and Office for National Statistics (ONS) mortality data (see Table S1 for code lists). Pre-specified secondary outcomes included hypotension, syncope, or fracture (defined identically) occurring within 1 and 5 years of the index date, to account for potential short-term clinical relevance.

### Model covariates
Potential predictors of hazardous hypotension, syncope, and fracture were identified based on published literature and consultation with clinical experts. A detailed summary of included variables is provided in Supplementary Table S2. A total of 40 predictors were assessed for the hypotension model, 41 for the syncope model, and 44 for the fracture model. These included demographic characteristics (such as age, sex, smoking history, and alcohol intake), medical history (e.g., prior relevant adverse events, diabetes, chronic kidney disease, stroke, atrial fibrillation, arrhythmias, osteoporosis, rheumatoid arthritis, epilepsy), and current medications (including but not limited to anti-hypertensives, opioids, sedatives, antidepressants, corticosteroids, and proton pump inhibitors; see Table S2). All comorbidities and clinical history were defined using relevant Read codes recorded any time before the index date. In contrast, medication exposure was

defined by at least one prescription issued in the 12 months prior to index.

### Sample size
A pre-specified sample size calculation was used to guide model development, yielding an estimated events-per-variable (EPV) range from 7 for the hypotension model to 20 for the fracture model. These estimates were based on assumptions of event rates between 18 and 51 per 10,000 person-years, a median follow-up duration of 7 years, an anticipated Nagelkerke's $R^2$ value of 0.15, and a maximum of 40 candidate predictor parameters per model[4,31]. Under these assumptions, the required number of outcome events was estimated to range from 277 to 784. The development cohort from CPRD GOLD substantially exceeded these requirements.

For external validation, the syncope model required a minimum sample of ~8000 individuals, including at least 400 events, to ensure a 95% confidence interval width of 0.2 around the estimated calibration slope[32]. This estimate was derived under several assumptions: a skew-normal distribution for the linear predictor with mean 0.16, variance 0.5, skewness 1, and kurtosis 4; an exponential survival time distribution with a baseline hazard rate of 0.008 (corresponding to 89% survival at 10 years); and an exponential distribution for censoring times with a rate of 0.2 (implying ~87% censoring by 10 years). Comparable sample size

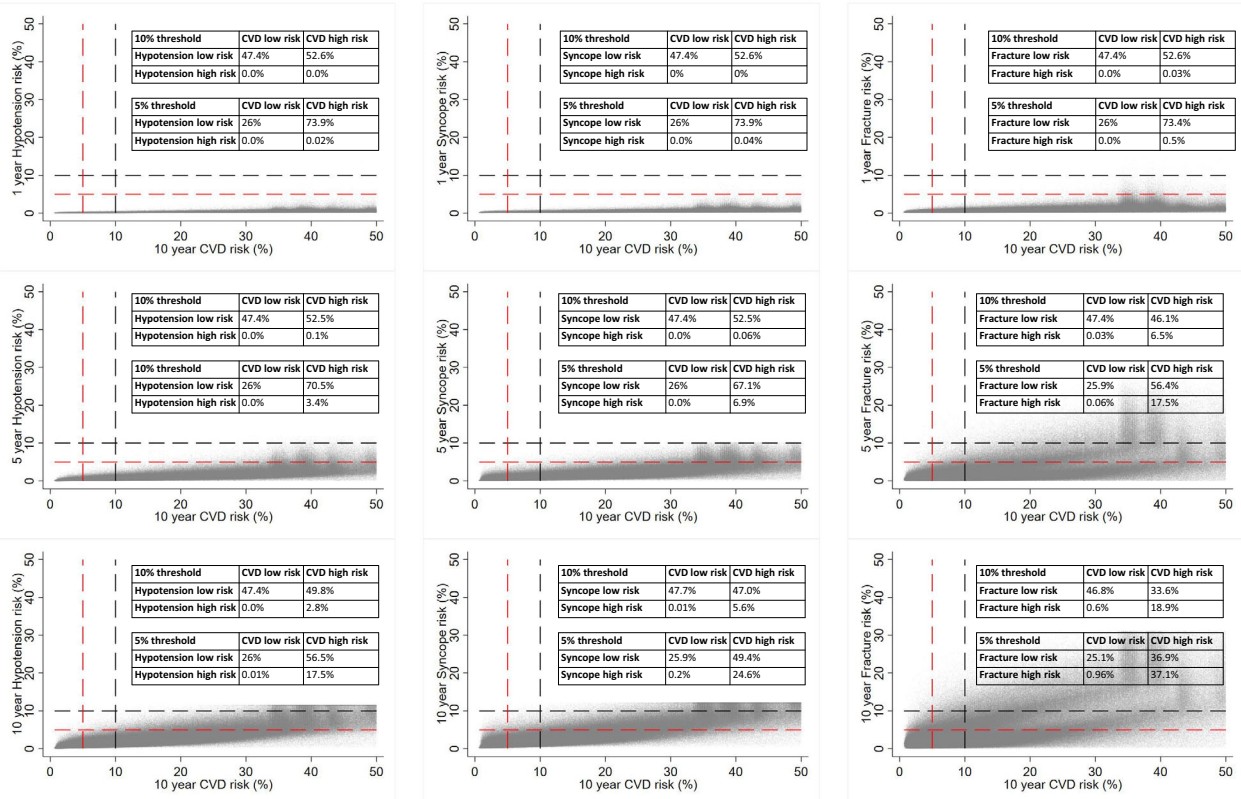

**Fig. 4 | Comparison of 10-Year Cardiovascular Risk (QRisk2) with STRATIFY Risk Scores (CPRD Gold).** Comparison of 10-year cardiovascular risk (QRisk2) and final STRATIFY-Hypotension, STRATIFY-Syncope and STRATIFY-Fracture risk in the CPRD Gold dataset (derivation). Red dashed lines indicate 5% risk threshold; Black dashed lines indicate 10% risk CVD cardiovascular disease.

estimates were obtained for the other outcomes. The CPRD Aurum validation dataset also exceeded these sample size thresholds.

## Statistical analysis

The analysis was conducted using the statistical software R versions 4.02 and 4.1.1 and STATA 16. All models are reported in line with the transparent reporting of a multivariable prediction model for individual prognosis or diagnosis (TRIPOD) guidelines for reporting of clinical prediction models (see Guideline S1 in appendix)[33]. Descriptive statistics were calculated for baseline characteristics in the model development and external validation cohorts separately.

## Model development

Model development and internal validation were carried out by researchers at the University of Oxford (CK, AW, JPS). For each imputed dataset, multivariable models were constructed using Fine-Grey subdistribution hazard regression to account for the competing risk of death from other causes[34]. This approach was selected to avoid overestimating the cumulative incidence of adverse events in the presence of competing mortality[35]. Model coefficients are presented as subdistribution hazard ratios (SHRs) with corresponding 95% confidence intervals. Baseline cumulative incidence functions were estimated post-estimation using a Breslow-type method as outlined by Fine and Grey[34]. Analyses were undertaken using the fastcmprsk package in RStudio[36]. Automated selection algorithms were not employed; all predictors were predefined based on prior literature and expert clinical judgement. Given the large sample size, most variables would be statistically significant, so further filtering was used to improve parsimony. Specifically, predictors with SHRs close to 1 (i.e., between 0.95 and 1.05) and low prevalence were excluded in the final model-fitting

stage. Shrinkage or penalisation methods to adjust for overfitting were not necessary due to the very large sample size.

To assess the linearity of continuous predictors (age, systolic and diastolic blood pressure, frailty index), fractional polynomial transformations were used[37]. The best-fitting transformation for each continuous covariate was applied uniformly across all imputations to ensure consistent coefficient estimates.

Potential interactions between age, sex, and antihypertensive therapy were explored but were excluded due to convergence issues, model instability and for the sake of parsimony. The proportional hazards assumption for each covariate was evaluated using Schoenfeld residuals[38].

## Apparent validation using development data

Apparent calibration of the models was evaluated using calibration plots that compared predicted and observed risks at 1, 5, and 10 years. Observed outcome probabilities were estimated using pseudo-values–jackknife-based estimators that quantify an individual's contribution to the cumulative incidence function for each outcome while accounting for the competing risk of death, derived using the Aalen–Johansen method[39]. To enhance stability, pseudo-values were computed separately within 50 groups stratified by linear predictor values. These calculations accounted for both competing risks and non-informative right censoring[40,41]. Calibration plots were constructed from the pseudo-values, incorporating a non-parametric smooth curve (symmetric nearest neighbour smoothing) with 95% confidence intervals to visualise the agreement between predicted and observed risks across the full risk spectrum[42]. Plots were generated separately for each imputed dataset, and consistency across imputations was assessed.

When miscalibration was observed at any time point (1, 5, or 10 years), the original model was recalibrated for that time point by fitting a generalised linear model with a logit link to the observed pseudo-values in the development cohort. The recalibration model used only the original model's linear predictor as the independent variable and allowed for non-linear recalibration via fractional polynomials. These recalibrated models were then subjected to external validation using the independent validation dataset.

## External validation

External validation of the prediction models was conducted independently by researchers at the University of Birmingham (LA, KIES, RDR), separate from the model development team. The full prediction algorithms are presented in the supplementary appendix (Equations S1–S3) and were applied to individuals in the validation dataset to generate predicted probabilities of hypotension, syncope, or fracture within 1, 5, and 10 years, accounting for the competing risk of death from other causes[43]. Calibration was assessed by comparing predicted risks to observed event probabilities, estimated using pseudo-values as described previously[39].

Model performance was summarised using calibration plots, observed-to-expected (O/E) ratios, Harrell's C-statistic, and Royston's D-statistic along with its associated $R^2$ each calculated using the pseudo-values as described above. To evaluate variability in model performance across general practices, we used random-effects meta-analysis with restricted maximum likelihood estimation (REML), acknowledging that case mix and event incidence may differ between sites[25,44]. The O/E ratio was pooled on the natural log scale, the C-statistic on the logit scale (standard errors derived via the delta method), and the D-statistic was pooled on its original scale[45,46]. Pooled estimates are reported with prediction intervals (PI) to give an indication of expected model performance in a new GP practice.

Clinical utility was examined using decision curve analysis for a range of potential threshold probabilities probabilities[47]. A decision threshold is defined as the probability at which a patient is classified as high risk and thus a decision has to be made (i.e., treat, refer for further investigation, etc.). The range of probabilities should reflect potential decision thresholds for the STRATIFY models. In the UK a CVD risk of 10% is considered high and given that the STRATIFY models should be used alongside a CVD risk tool a range of threshold probabilities of up to 20% was considered reasonable. The final models for each outcome were compared at 1, 5 and 10 years to 'model-blind methods' of (a) introducing adverse event prevention measures (which may include deprescribing) for all patients or (b) not introducing adverse event prevention measures for all patients, regardless of risk. If the models have higher net benefit than the strategies (a) or (b) then this would suggest using the models to inform prescribing would be preferable.

Direct comparison with existing cardiovascular risk prediction tools (e.g., QRisk2) using decision curve analysis is not possible. Therefore, to further explore clinical utility and potential implementation, we examined the relationship between predicted risk of hypotension, syncope, and fracture (at 1, 5, and 10 years) and 10-year cardiovascular risk using QRisk2 at 5% and 10% thresholds[9]. The overlap in patients identified as high-risk ($\geq 5\%$ and $\geq 10\%$) by the STRATIFY models versus QRisk2 was quantified and visualised using Venn diagrams.

## Missing data

To address missing data in both the development and validation cohorts, we employed multiple imputation using chained equations, generating ten imputed datasets for each cohort. Separate imputation procedures were conducted independently for the development and validation datasets. The imputation models included all model covariates within each dataset, along with the Nelson-Aalen estimator for the cumulative baseline cause-specific hazards for hypotension, syncope or fracture and for the competing event of death, and binary event indicators for each of these possible event types[48,49]. For comorbidity diagnoses and prescribed medications, missingness was handled under the assumption that absence of data reflected absence of diagnosis or prescription. Variables imputed included ethnicity, body mass index (BMI) category, smoking status, alcohol consumption, and (in the validation cohort only) the deprivation score.

Imputations were assessed for consistency and validity by comparing density plots, histograms, and summary statistics across imputations and back to complete values. Following imputation, model coefficients and performance metrics were estimated separately within each imputed dataset and then combined using Rubin's Rules[50]. In instances where Rubin's Rules were inappropriate due to non-normal posterior distributions, summary measures across imputations were reported using the median and interquartile range (IQR)[51]. A sensitivity analysis using a full case approach was also employed to compare with the imputed models.

## Reporting summary
Further information on research design is available in the Nature Portfolio Reporting Summary linked to this article.

## Data availability
This study is based on data obtained through an institutional license from the Clinical Practice Research Datalink (CPRD). Access to these data is restricted and permitted only to researchers with approved study protocols reviewed by CPRD's Independent Scientific Advisory Committee (ISAC). Data access may involve a fee and is subject to specific governance and licensing conditions. Full details on how to request access, including contact information and procedures, are available at: (https://www.cprd.com/data-access).

## Code availability
The developed algorithms are freely available for research use and can be downloaded from (https://process.innovation.ox.ac.uk/software/). Code lists used to define variables included in the dataset are available at (https://github.com/jamessheppard48/STRATIFY-BP (https://doi.org/10.5281/zenodo.15481343).

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

## Acknowledgements

We thank Lucy Curtin for administrative support throughout the project, and Simon Griffin and Juliet A Usher-Smith (Department of Public Health and Primary Care, Primary Care Unit, University of Cambridge) for their contributions as STRATIFY Investigators to the project. We thank Lucy Curtin for administrative support throughout the project, and Simon Griffin and Juliet A Usher-Smith (Department of Public Health and Primary Care, Primary Care Unit, University of Cambridge) for their contributions as STRATIFY Investigators to the project. The Hospital Episode Statistics data used in this analysis are re-used with permission from NHS Digital who retain the copyright for those data. We thank the Office for National Statistics for providing data on mortality. The Office for National Statistics, and NHS Digital bear no responsibility for the analysis or interpretation of the data. Finally, we are very grateful to all those patients who permit their anonymized routine NHS data to be used for this approved research. For this research, JPS and CK were funded in whole, or in part, by the Wellcome Trust/Royal Society via a Sir Henry Dale Fellowship held by JPS (ref: 211182/Z/18/Z) and the National Institute for Health Research (NIHR) School for Primary Care (project 430). LA, KIES and RR are supported by funding from the NIHR Birmingham Biomedical Research Centre at the University Hospitals Birmingham NHS Foundation Trust and the University of Birmingham. CEC is part supported by a NIHR School for Primary Care grant (project 580). RJMcM is supported by an NIHR Senior Investigator award and by NIHR ARC Oxford Thames Valley. FDRH acknowledges part support from the NIHR ARC Oxford Thames Valley, and the NIHR Oxford OUH BRC. KIES is funded by an NIHR SPCR Launching Fellowship. AB has received research funding from AstraZeneca, NIHR, BMA Medical Research Foundation and UKRI. RAP receives funding from the NIHR. AC is part funded by NIHR ARC Yorkshire & Humber and Health Data Research UK, an initiative funded by UK Research and Innovation Councils, National Institute for Health Research and the UK devolved administrations, and leading medical research charities. The views expressed are those of the author(s) and not necessarily those of the NHS, the NIHR or the Department of Health and Social Care. For the purpose of Open Access, the author has applied a CC BY public copyright licence to any Author Accepted Manuscript version arising from this submission. The sponsor and funders had no role in the design and conduct of the study; collection, management, analysis, and interpretation of the data; preparation, review, or approval of the manuscript; and decision to submit the manuscript for publication.

## Author contributions

J.P.S. conceived the project and wrote the protocol with F.D.R.H., R.J.M., R.J.S., and R.D.R. C.K., S.S. and A.W. extracted data for analysis. CK and AW developed the models under supervision of JS and RS; FDRH, CEC, RJM, AC, AB and RAP advised on model development. LA validated the model under supervision of RDR and KIES. JS wrote the first draft of the manuscript. MO contributed as the Patient and Public Involvement (PPI) representative. All authors revised the manuscript and approved the final version. JPS is the guarantor for this work and accepts full responsibility for the conduct of the study, had access to the data, and controlled the decision to publish. The principal investigator (JPS) attests that all listed authors meet authorship criteria and that no others meeting the criteria have been omitted.

## Competing interests

All authors have completed the ICMJE uniform disclosure form at (www.icmje.org/coi_disclosure.pdf) and declare: authors had financial support from the Wellcome Trust, Royal Society, and National Institute for Health Research for the submitted work; no financial relationships with any organisations that might have an interest in the submitted work in the previous three years; no other relationships or activities that could appear to have influenced the submitted work.

## Additional information

## the STRATIFY investigators

Simon Griffin[10] & Juliet A. Usher-Smith[10]

[10]Department of Public Health and Primary Care, Primary Care Unit, University of Cambridge, Cambridge, UK. A full list of members and their affiliations appears in the Supplementary Information.

