## [Transparent Peer Review file · Nature Communications]

Predicting the risk of adverse events in patients with an indication for antihypertensive treatment: development and external validation of the STRATIFY-Hypotension, STRATIFY-Syncope and STRATIFY-Fracture prediction models

Corresponding Author: Dr Constantinos Koshiaris

Version 0:

Reviewer comments:

Reviewer #1

(Remarks to the Author)

- Developed models help to identify risk factors for hypotension, syncope and fractures in treated hypertensive patients and globally reassure about the relatively low risk of these outcomes from antihypertensive therapy.
- The work is significant in the specific field and adds further knowledge.
- The work supports the conclusions and claims.
- I did not find flaws in the data analysis.
- The methodology is sound and meets the expected standards in the field.
- There is enough detail in the methods.

Summary

Antihypertensive therapy may be associated with increased risk of syncope, hypotension, and fractures, but the highest-risk individuals are unclear. This study developed and validated three models to predict these outcomes in patients with an indication for antihypertensive therapy. A cohort study was conducted using data from Clinical Practice Research Datalink (CPRD). Patients aged > 40 years with systolic BP 130-179 mmHg were included. Outcomes were first hypotension, syncope, or fracture leading to hospitalization or death within 10 years. Models were derived from CPRD GOLD data (n = 1,773,224) and validated with CPRD Aurum data (n = 3,805,366). Each model had 31-38 predictors. Validation showed excellent discrimination (C-statistic at 10 years: Hypotension 0.822; Syncope 0.817; Fracture 0.790) and acceptable calibration. The authors conclude that these models could be used to help reassure patients about the relatively low risk of harm from antihypertensive therapy, or identify the small number of individuals with a higher risk where additional monitoring may be indicated.

General comment

This is an interesting and well written manuscript. Developed models help to identify risk factors for hypotension, syncope and fractures in treated hypertensive patients and globally reassure about the relatively low risk of these outcomes from antihypertensive therapy. I have only a few comments.

Specific Comments

“Strong predictors of fracture included heavy drinking, female sex, chronic liver disease, previous fracture, multiple sclerosis, and epilepsy. All antihypertensive medications had a weak or no association with the risk of fracture. Other medications were associated with an increased risk of fracture, with the exception of antiandrogen therapy which conferred a lower risk of fracture (table 2).” Could osteoporosis and rheumatoid arthritis be included? Are there data about antiestrogen therapy? “Subgroup analyses of the 10-year risk models showed similar performance in younger (<65 years) and older patients (≥65 years) and in females and males (figures S8, S10; tables S6 and S7).” I suggest to replace “younger” with “adult” and “older” with “old”.

There are some typos.

Reviewer #2

(Remarks to the Author)

The study uses very large dataset from EHR in UK to derive new prediction models for hypotension, syncope, and fracture. I think the analytic approach is generally valid, but mainly I have issues about the aim/study design and interpretations of the results.

Major:

- Most importantly, the models are developed for a clinical usage to decide indication of antihypertensives, but the outcomes are not necessarily caused by antihypertensives, which is inconsistent. In other words, clinicians want to know the risk of hypotension due to initiation or modification of dosages of antihypertensives. Currently, the model may predict orthostatic hypotension or shock. The outcomes should be defined aligning to their intended clinical usage.
- Also, the predicted risk is not like 'the risk if antihypertensives are initiated.' Even if the baseline risk is high, the model does not tell whether the patients will have increased risk due to antihypertensives. But this is what we want to know. The analytic approach for this would be different from the present conventional (factual) prediction, so may be beyond the scope of this paper. But then the interpretation of the results should be more conservative, as it does not indicate clinical utilities. Current clinical utility analysis is not causal inference (i.e. g-method as this is a time-varying exposure) as I read – if it is not correct, please provide the details
- Provide ROC curves or sensitivity/specificity. c-statistic of 0.82 is not necessarily regarded as high since the outcome is rare. Just predicting 'no event' for everyone can lead to relatively good prediction

Minor:

- External validation is for risk. Then it would be better to develop prediction models for risk too, not only time-to-event
- I do not think the calibration was "acceptable". Please provide reasoning.

Version 1:

Reviewer comments:

Reviewer #1

(Remarks to the Author)

The authors have answered my questions. I have no further comments.

Reviewer #2

(Remarks to the Author)

I thank authors for incorporating my comments. The manuscript has been improved and I think it is now acceptable for the journal.

Two remaining minor comments:

- Add interpretation of survival AUC in abstract and Results. Unless readers may misinterpret the results as usual AUC value. I personally want to know how to interpret the value.
- O/E ratio is significantly off from 1. STRATIFY-Fracture has the estimate of 1.13. I do not think this shows 'excellent' calibration, but the model underestimates risk or the hospital performs worse than expected. More nuanced interpretation would be necessary.

Version 2:

Reviewer comments:

Reviewer #2

(Remarks to the Author)

Thank you for the edits. I hope that this manuscript will be well read.

We sincerely thank the reviewers for their thoughtful and constructive feedback. In response to the reviewer comments, we re-ran our analyses and identified an issue with the outcome definition in the validation dataset, which contributed to the observed miscalibration in the external dataset. After correcting this issue and repeating the analysis, we observed a substantial improvement in calibration for nearly all models. We have highlighted all changes made in both the manuscript and the appendix. Below, we address each comment point by point.

REVIEWER COMMENTS

Reviewer #1 (Remarks to the Author):

- Developed models help to identify risk factors for hypotension, syncope and fractures in treated hypertensive patients and globally reassure about the relatively low risk of these outcomes from antihypertensive therapy.**
- The work is significant in the specific field and adds further knowledge.**
- The work supports the conclusions and claims.**
- I did not find flaws in the data analysis.**
- The methodology is sound and meets the expected standards in the field.**
- There is enough detail in the methods.**

Summary

Antihypertensive therapy may be associated with increased risk of syncope, hypotension, and fractures, but the highest-risk individuals are unclear. This study developed and validated three models to predict these outcomes in patients with an indication for antihypertensive therapy. A cohort study was conducted using data from Clinical Practice Research Datalink (CPRD). Patients aged > 40 years with systolic BP 130-179 mmHg were included. Outcomes were first hypotension, syncope, or fracture leading to hospitalization or death within 10 years. Models were derived from CPRD GOLD data (n = 1,773,224) and validated with CPRD Aurum data (n = 3,805,366). Each model had 31-38 predictors. Validation showed excellent discrimination (C-statistic at 10 years: Hypotension 0.822; Syncope 0.817; Fracture 0.790) and acceptable calibration. The authors conclude that these models could be used to help reassure patients about the relatively low risk of harm from antihypertensive therapy, or identify the small number of individuals with a higher risk where additional monitoring may be indicated.

General comment

This is an interesting and well written manuscript. Developed models help to identify risk factors for hypotension, syncope and fractures in treated hypertensive patients and globally reassure about the relatively low risk of these outcomes from antihypertensive therapy. I have only a few comments.

Specific Comments

“Strong predictors of fracture included heavy drinking, female sex, chronic liver disease, previous fracture, multiple sclerosis, and epilepsy. All antihypertensive medications had a weak or no association with the risk of fracture. Other medications were associated with an increased risk of fracture, with the exception of antiandrogen therapy which conferred a lower risk of fracture (table 2).” Could osteoporosis and rheumatoid arthritis be included? Are there data about antiestrogen

therapy?

“Subgroup analyses of the 10-year risk models showed similar performance in younger (<65 years) and older patients (≥65 years) and in females and males (figures S8, S10; tables S6 and S7).” I suggest to replace “younger” with “adult” and “older” with “old”.

There are some typos.

We thank the reviewer for their thoughtful and positive comments. Osteoporosis and rheumatoid arthritis were indeed included in the final model, as shown in Table 2. Osteoporosis had a SHR of 1.30 (95% CI: 1.25–1.34), and rheumatoid arthritis had a SHR of 1.30 (95% CI: 1.24–1.37). For further clarity, we have updated the results section as follows:

Line 269–271:

'Strong predictors of fracture included heavy drinking, female sex, chronic liver disease, previous fracture, multiple sclerosis, epilepsy, osteoporosis, and rheumatoid arthritis.'

Regarding hormone therapy types, we included Hormone Replacement Therapy (HRT), which we identified from the literature as being associated with fractures (Appendix Table S2). We did not extract data related to antiestrogen therapy or other hormone-blocking therapies, and unfortunately it is not possible to do so at this stage. We have also corrected the variable name, as it was previously reported incorrectly and rather than referring to hormone replacement therapy it was reported as hormone blocking therapy.

We have thoroughly reviewed the manuscript for typos and have addressed those we detected. After discussing the terminology with our patient representative, we have decided to retain 'younger' and 'older' in the manuscript, as the term 'old' can have negative connotations, particularly in the UK.

Reviewer #2 (Remarks to the Author):

The study uses very large dataset from EHR in UK to derive new prediction models for hypotension, syncope, and fracture. I think the analytic approach is generally valid, but mainly I have issues about the aim/study design and interpretations of the results.

Major:

- **Most importantly, the models are developed for a clinical usage to decide indication of antihypertensives, but the outcomes are not necessarily caused by antihypertensives, which is inconsistent. In other words, clinicians want to know the risk of hypotension due to initiation or modification of dosages of antihypertensives. Currently, the model may predict orthostatic hypotension or shock. The outcomes should be defined aligning to their intended clinical usage.**

We thank the reviewer for this insightful comment and agree with their concern. Our models do not imply that antihypertensive medications cause these outcomes; rather it estimates an individual's baseline risk (with or without antihypertensive treatment) of being hospitalized or dying from these specific outcomes. This is an essential first step because, to assess the impact of initiating or modifying

antihypertensive treatment, clinicians need to understand a patient's baseline risk of adverse events, independent of treatment status. We included antihypertensive treatment in the model to ensure that patients already on treatment, who may benefit from modifications, were not excluded. However, we do not imply that this represents a causal effect. The utility of this model lies in its ability to provide a baseline risk estimate. To determine how treatment modification alters this risk, our estimates must be integrated with evidence from randomized controlled trials or high-quality observational studies (1,2). Because this study focuses solely on baseline risk prediction, its findings should not be interpreted as causal effects of antihypertensive treatment. We have been careful with the statistical terminology we have used to avoid suggesting causation. In addition, we have clarified this distinction in the manuscript by adding the following:

- End of introduction section (lines 101-105): The present study therefore aimed to develop and externally validate three new prediction models for adverse events commonly associated with antihypertensive treatment—namely hypotension, syncope, and fracture—that have significant impacts on individuals and health care systems, with the goal of estimating the baseline risk of these adverse reactions, regardless of whether the patients are on antihypertensive medication.
 - We replaced with estimation of “baseline risk” where appropriate.
 - Discussion (lines 377-382): The purpose of developing these risk prediction models was to help clinicians estimate the baseline risk of adverse reactions. The models can be applied to both patients that are on antihypertensive treatment or patients for which treatment is being considered. Clinicians can then combine the estimated baseline risk with relative treatment effects obtained from well conducted randomized clinical trials or observational studies to estimate how the risk gets modified by starting, changing or altering the dosage of a medication [3,4]. Based on this updated risk different treatment strategies can be considered depending on the outcome of interest. For example, in patients at high risk of hypotension and fracture, modification of antihypertensive treatment may be considered, whereas in patients at risk of fracture alone, other prevention strategies may be more appropriate. These tools should be used alongside CVD risk estimation tools to get a more complete picture of the harm/benefit profile of the patient. This can enable better informed decisions regarding when to prescribe, continue or deprescribe antihypertensive treatment. Lines 396-398: Changed to “For most patients, these models should be employed to provide reassurance that the risk of adverse events is low for the vast majority of the population even after taking into consideration treatment effects from other studies”
- 1) Albasri A, Hattle M, Koshiaris C, Dunnigan A, Paxton B, Fox SE, et al. Association between antihypertensive treatment and adverse events: systematic review and meta-analysis. *BMJ* 2021;372. <https://doi.org/10.1136/BMJ.N189>.
 - 2) Sheppard JP, Koshiaris C, Stevens R, Lay-Flurrie S, Banerjee A, Bellows BK, et al. The association between antihypertensive treatment and serious adverse events by age and frailty: A cohort study. *PLoS Med* 2023;20. <https://doi.org/10.1371/JOURNAL.PMED.1004223>

• Also, the predicted risk is not like 'the risk if antihypertensives are initiated.' Even if the baseline risk is high, the model does not tell whether the patients will have increased risk due to antihypertensives. But this is what we want to know. The analytic approach for this would be different from the present conventional (factual) prediction, so may be beyond the scope of this paper. But then the

interpretation of the results should be more conservative, as it does not indicate clinical utilities. Current clinical utility analysis is not causal inference (i.e. g-method as this is a time-varying exposure) as I read – if it is not correct, please provide the details

We thank the reviewer for this insightful comment and fully agree. As stated, this is not a causal inference study but one aimed at estimating the baseline risk of adverse side effects, upon which treatment effects can be applied. Incorporating treatment effects is beyond the scope of this study, which primarily focuses on baseline risk estimation. We have previously published a causal inference paper that examines the association between antihypertensive medication and side effects (references provided in previous reply), providing a true estimate of treatment effects.

The clinical utility analysis, particularly the Decision Curve Analysis (DCA), is a key methodological aspect of prediction modeling. DCA evaluates the net benefit of using a model's predictions compared to other approaches across various thresholds. It is based on factual predictions, and we acknowledge the limitation that it does not account for causal treatment effects. Specifically, the predicted risks reflect associational relationships, not counterfactual risks (i.e., "What would the patient's risk be if we were to change or add a medication?").

Nevertheless, a model with a higher net benefit indicates improved decision-making, as it more effectively identifies high-risk patients for intervention and reduces unnecessary treatments for low-risk patients. Although this does not directly evaluate the causal impact of treatment, we believe that a model with higher net benefit would lead to better clinical outcomes in practice, as improved risk stratification allows for more targeted interventions such as intensification, removal, or deprescribing of treatments.

To address this point in the manuscript, we have moderated our conclusions about clinical utility and added the following clarification in the strengths and limitations section (Lines 407-417):

"Findings from clinical utility analyses should be interpreted with caution, as they estimate factual risk (i.e., risk based on baseline characteristics). The net benefit in Decision Curve Analysis (DCA) reflects the clinical utility of using the model's predictions for treatment decision-making, without accounting for the causal effect of altering antihypertensive medication on outcomes. This study primarily aims to predict baseline risk and does not evaluate how treatment changes will modify these risks.

While the model enhances risk stratification and supports decision-making by identifying high- and low-risk patients, it does not provide direct insight into how starting or adjusting antihypertensive treatments affects patient outcomes. Thus, any conclusions regarding treatment effects should rely on additional evidence, such as randomized controlled trials or observational studies. However, it is reasonable to assume that a model with higher net benefit may still improve clinical outcomes by better targeting interventions"

• Provide ROC curves or sensitivity/specificity. c-statistic of 0.82 is not necessarily regarded as high since the outcome is rare. Just predicting 'no event' for everyone can lead to relatively good prediction

We thank the reviewer for the comment. However, we believe that ROC curves and traditional metrics such as sensitivity and specificity are not suitable for time-to-event data, particularly due to censoring.

These traditional methods assess binary outcomes without accounting for time or censoring, making them inappropriate for our analysis. Instead, we used Harrell's C-statistic, which is specifically designed for time-to-event data and evaluates the model's ability to differentiate between subjects based on their risk of experiencing the event over time. This approach is conceptually similar to the ROC curve in binary classification tasks.

We also acknowledge the concern about the potential inflation of the C-statistic (or other similar measures) by predicting 'no event.' To address this, we have focused on Decision Curve Analysis (DCA), which evaluates the "clinical utility" of the model by considering its potential benefits in real-world decision-making. DCA estimates the net benefit of using a model compared to other strategies and goes beyond discrimination by assessing whether the model is valuable in guiding practical decisions. Net benefit refers to the balance between the correct predictions that lead to beneficial outcomes (true positives) and the incorrect predictions (false positives or false negatives) that may result in unnecessary interventions or missed opportunities, helping to assess the overall value of the model in decision-making (approach and methodology differ for time-to-event and survival data due to censoring, the underlying principle remains the same).

It has become an essential tool in prediction modeling due to the limitations of statistics like AUC. One of the reasons we use DCA is that a model with a "good" AUC might not necessarily provide practical value when tested in real-world decision-making scenarios. This is evident from the DCA analysis of the hypotension model at 1-year. While that C-statistic is high (0.82) the DCA shows that there is no clinical utility as the net benefit is lower than other strategies which is why we suggest caution when using the 1-year version.

Minor:

- **External validation is for risk. Then it would be better to develop prediction models for risk too, not only time-to-event**

We thank the reviewer for their comment. However, we are not entirely clear on the specific point being raised, as our models already predict the risk of the event occurring. Specifically, the survival models we developed estimate the predicted probability (cumulative incidence/risk) of the event occurring at 1, 5, or 10 years (equations provided in appendix S1, S2, S3). The model equations were then applied to the external validation dataset to obtain the risk for the patients in that dataset at 1, 5 and 10 years. Using these predicted probabilities, we evaluated the performance of the model in the external dataset in terms of calibration, discrimination, and clinical utility

- **I do not think the calibration was "acceptable". Please provide reasoning.**

We thank the reviewer for their comment. To ensure that no analysis-related issues were affecting the calibration, we conducted a thorough review at both the derivation and validation stages. During this process, we identified an error in defining the outcome variables in the validation dataset for the syncope and hypotension models, which impacted the model's calibration. After correcting this issue and repeating the analysis, we observed a significant improvement in calibration in almost all models and now they perform exceptionally well. The only one that is still not performing well is the 1-year hypotension model which we acknowledge in the discussion. We have also updated all relevant figures,

tables, and appendix materials in the manuscript to reflect these improvements. In addition, we added the following for further clarification:

Discussion (Lines 327-328): Calibration was excellent for all models and across all time horizons with the exception of the hypotension model at 1 year which severely underestimates the risk. Given this, we would suggest caution in using the 1-year hypotension model at this stage

Reviewer replies

Reviewer #1 (Remarks to the Author):

The authors have answered my questions. I have no further comments.

We thank the reviewer for their feedback. We appreciate their time and consideration, and we're glad to hear that our responses addressed the questions.

Reviewer #2 (Remarks to the Author):

I thank authors for incorporating my comments. The manuscript has been improved and I think it is now acceptable for the journal.

We thank the reviewer for their thoughtful feedback. We are pleased to hear that the manuscript has been improved. We appreciate your two remaining minor comments and will ensure they are thoroughly addressed in the revised version.

Two remaining minor comments:

- Add interpretation of survival AUC in abstract and Results. Unless readers may misinterpret the results as usual AUC value. I personally want to know how to interpret the value.

We thank the reviewer for the comment and we provide further clarification on the C-statistic. Harrell's C-statistic as reported in our study is interpreted similarly to the standard AUC: a value of 0.5 indicates no discrimination, while values approaching 1.0 suggest better discrimination. The key difference lies in its estimation, as the C-statistic accounts for censoring, preventing the overestimation of discrimination that may occur with standard AUC. While Harrell's C-statistic shares the same interpretation as AUC, its values may be lower in survival analysis due to censored data. This provides a more conservative but realistic measure of model discrimination.

To address the reviewer's concern, we have included a clarification on the interpretation of the C-statistic in the Methods section. Due to word count limitations and to maintain the abstract's focus on key findings, we did not include this explanation there.

"Predictive performance was quantified using calibration plots, and by calculating the observed to expected ratio (O/E), Harrell's C-statistic (measure of discrimination for survival data interpreted similarly to the AUC statistic), and Royston's D-statistic with its associated R2 statistic."

- O/E ratio is significantly off from 1. STRATIFY-Fracture has the estimate of 1.13. I do not think this shows 'excellent' calibration, but the model underestimates risk or the hospital performs worse than expected. More nuanced interpretation would be necessary.

We thank the reviewer for the comment. While the O/E ratio of 1.13 may seem distant from 1, it should be interpreted in conjunction with the calibration plot, as individual statistics often don't tell the full story. This is similar to how we approach discrimination statistics, which is why we also emphasize decision curve analysis. Upon reviewing the calibration plot, we observe that the primary issue arises for very low predicted probabilities (below 3%). For example, a predicted probability of 2.5% corresponds to

an observed probability of approximately 2.8%, indicating that the observed is about 13% higher—though this difference is small in absolute terms.

Given that many of our predicted probabilities are below 5% (as shown in the histogram below the calibration plot), the O/E ratio is somewhat skewed towards showing underprediction. However, this difference is unlikely to significantly impact decision-making, particularly when decision thresholds are above 5%, where the model performs well.

We have adjusted our conclusion to acknowledge the slight underprediction observed for low probabilities. Specifically, we updated the following sections to reflect this nuance:

Abstract:

Validation showed excellent discrimination (C-statistic at 10 years: Hypotension 0.824; Syncope 0.819; Fracture 0.790) and excellent calibration for the hypotension and syncope model though some underprediction was observed for the fracture model

Results:

Lines 285-287: The STRATIFY-Fracture model showed good discrimination (C-statistic at 10 years 0.790, 95% CI 0.789 to 0.792) and good calibration but with some underprediction for low probabilities (O/E ratio at 10 years 1.13, 95% CI 1.11 to 1.14).

Discussion:

Calibration was excellent for all models across all time horizons, with the exception of the hypotension model at 1 year, which tends to underestimate the risk. Some minor underprediction was also observed for the fracture model at 10 years for low predicted probabilities.